# Epigallocatechin Gallate (EGCG), an Active Phenolic Compound of Green Tea, Inhibits Tumor Growth of Head and Neck Cancer Cells by Targeting DNA Hypermethylation

**DOI:** 10.3390/biomedicines11030789

**Published:** 2023-03-05

**Authors:** Anshu Agarwal, Vikash Kansal, Humaira Farooqi, Ram Prasad, Vijay Kumar Singh

**Affiliations:** 1Department of Zoology, Agra College, Dr. Bhimrao Ambedkar University, Agra 282004, India; 2Department of Otolaryngology, Emory University, Atlanta, GA 30322, USA; 3Department of Biochemistry, Hamdard University, New Delhi 110062, India; 4Department of Ophthalmology and Visual Sciences, University of Alabama at Birmingham, Birmingham, AL 35294, USA; 5Narain PG Degree College, Shikohabad, Dr. Bhimrao Ambedkar University, Agra 282004, India

**Keywords:** epigenetics, DNA methylation, growth, HNSCC, EGCG

## Abstract

Head and neck cancers are among the deadliest cancers, ranked sixth globally in rates of high mortality and poor patient prognoses. The prevalence of head and neck squamous cell carcinoma (HNSCC) is associated with smoking and excessive alcohol consumption. Despite several advances in diagnostic and interventional methods, the morbidity of subjects with HNSCC has remained unchanged over the last 30 years. Epigenetic alterations, such as DNA hypermethylation, are commonly associated with several cancers, including HNSCC. Thus, epigenetic changes are considered promising therapeutic targets for chemoprevention. Here, we investigated the effect of EGCG on DNA hypermethylation and the growth of HNSCC. First, we assessed the expression levels of global DNA methylation in HNSCC cells (FaDu and SCC-1) and observed enhanced methylation levels compared with normal human bronchial epithelial cells (NHBE). Treatment of EGCG to HNSCC cells significantly inhibited global DNA hypermethylation by up to 70–80% after 6 days. Inhibition of DNA hypermethylation in HNSCC cells was confirmed by the conversion of 5-methylcytosine (5-mc) into 5-hydroxy methylcytosine (5hmC). DNA methyltransferases regulate DNA methylation. Next, we checked the effect of EGCG on the expression levels of DNA methyltransferases (DNMTs) and DNMT activity. Treatment of EGCG to HNSCC cells significantly reduced DNMT activity to 60% in SCC-1 and 80% in FaDu cells. The protein levels of DNMT3a and DNMT3b were downregulated in both cell lines after EGCG treatment. EGCG treatment to HNSCC cells reactivated tumor suppressors and caused decreased cell proliferation. Our in vivo study demonstrated that administration of EGCG (0.5%, *w*/*w*) as a supplement within an AIN76A diet resulted in inhibition of tumor growth in FaDu xenografts in nude mice (80%; *p* < 0.01) compared with non-EGCG-treated controls. The growth inhibitory effect of dietary EGCG on the HNSCC xenograft tumors was associated with the inhibition of DNMTs and reactivation of silenced tumor suppressors. Together, our study provides evidence that EGCG acts as a DNA demethylating agent and can reactivate epigenetically silenced tumor suppressors to inhibit the growth of HNSCC cells.

## 1. Introduction

Head and neck squamous cell carcinoma (HNSCC) is the most common form of cancer worldwide and is associated with poor survival when diagnosed in the advanced stage [1]. A recent report published by the American Cancer Society estimated approximately 54,540 new cases of HNSCC and 11,580 deaths in the United States in 2023 [2]. The annual incidence of HNSCC is increasing at an alarming rate and has been estimated at more than 660,000 new cases and about 325,000 deaths annually around the world [3]. The increasing incidence of HNSCC is about 2.5 times higher in men (39,290) than in women (15,250) [2]. Collectively, HNSCC describes cancers of the oral cavity, pharynx, and larynx. The occurrence of HNSCC is linked with certain environmental and lifestyle risk factors, including tobacco and alcohol consumption. Due to intense habits of tobacco consumption, ~57.5% of total global cases of HNSCC are found in Asia, including 30% of all cancers in India [4]. The global economic impact of HNSCC is $535 billion dollars, while the cost to South Asia is $133 billion dollars [5]. Despite advancements in diagnostics and interventional methods, the discovery of newer biomarkers, molecular signaling, screening strategies, and anti-cancer drugs is still desirable as the 5-year survival for patients with HNSCC remains unchanged. Studies on naturally active biomolecules have indicated their anti-cancer and anti-inflammatory properties, and other chemopreventive activities, with fewer side effects. Therefore, investigating dietary interventions in managing cancers, including HNSCC, is important for public health.

Emerging evidence has revealed that phenolic compounds can modulate cells’ fates by altering molecular signaling. Epigallocatechin 3-gallate (EGCG) is an active phenolic compound of green tea. It has been studied for its antioxidant, anti-inflammatory, anti-tumor, and anti-carcinogenic properties by targeting molecular signaling, including autophagy, AMPK, eNOS, NF-kB, epigenetic modifications such as DNA methylation, and histone modifications [6,7,8,9]. In the pathogenesis of cancer, several mutations occur at the molecular level that change the fates of cells, eventually leading to uncontrolled cell division. During the development and maintenance of normal physiology, cell division is regulated by multiple cell cycle regulatory proteins such as p16^INK4A^, p21^Waf1/Cip1^, p27^Kip1^, p53, cyclin E, cyclin-dependent kinases (CDKs), etc., through genetic and epigenetic changes [10]. DNA methylation is one of the key epigenetic modifications contributing to mammalian development. DNA methylation occurs when a methyl group is added to the 5th position of cytosine rings located in CpG dinucleotides and converted into 5-methylcytosine (5mC) [11]. This primary epigenetic event is regulated by a family of enzymes named DNA methyltransferases (DNMTs) [12,13,14]. Although epigenetic processes are natural and essential for many functions, they can also cause significant adverse health effects if they are disrupted. Recent studies on epigenetics highlighted the role of DNA hypermethylation and demonstrated that the hypermethylation of CpG islands in the promoter regions of tumor suppressor genes causes gene silencing that contributes to the development and progression of cancer [15,16,17,18,19].

Although EGCG has been shown to inhibit DNA methylation and reversal of epigenetic changes in other cancers, the role of EGCG in HNSCC management remains unexplored. In this study, we investigated whether EGCG would reactivate silenced tumor suppressor genes, and determined the molecular mechanism underlying these effects, using two HNSCC cell lines derived from different sub-sites, such as pharynx and oral cavity, with in vitro and in vivo models.

## 2. Material and Methods

### 2.1. Chemicals and Antibodies

The purified EGCG (95%) was purchased from the commercial herb supplier, Maysar Herbals (Faridabad, Haryana 121004, India). The antibodies specific for p16 (#sc-1661), p21 (#sc-5246), p27 (#sc-1641), β-actin(#sc-47778), DNMT1 (#sc-271729), DNMT3a (#sc-365769), DNMT3b (#sc-376043), histone H3 (#sc-5661), 5-Aza-2-cytidine (5AzaDC; #sc-221003) from Santa Cruz Biotechnology (Santa Cruz, CA, USA), and 5-Methycytosine (5-mC, #28692) and 5-Hydroxymethylcytosine (5hmC, #51660) from Cell Signaling (Danvers, MA, USA), were purchased through an Indian supplier (New Delhi, India). Other analytical grade chemicals were purchased from Merck Ltd. (Mumbai, India) and Sigma-Aldrich (Bengaluru, India). Cell culture medium, fetal bovine serum (FBS), global DNA methylation (#P-1030), and DNMT activity assay kit (#P-3009) were purchased from Thermo Fisher Scientific India Pvt. Ltd., Mumbai, India.

### 2.2. HNSCC Cell Lines and Cell Culture Conditions

The HNSCC cell line UM-SCC-1 of the oral cavity (#SCC070) and normal human bronchial epithelial (NHBE, #C-14063) cells were purchased from Sigma-Aldrich (Bengaluru, India). The FaDu cells (hypopharyngeal carcinoma) were generously gifted. No further authentication was carried out by the authors. Cells were cultured and maintained in Dulbecco’s modified Eagle’s medium (DMEM) supplemented with 10% fetal bovine serum (FBS) and 1% penicillin/streptomycin (PS) in an incubator equipped with a humidifier to maintain atmosphere humidity 95%, 5% CO_2,_ at 37 °C. For the treatment purpose, cells were seeded in a T-75 flask and allowed to attach for 24 h before treatment with testing agents (3 or 6 days). Media and treatment agents were refreshed every day. About 60–70% of confluent cells were treated with either varying concentrations of EGCG (0, 5, 10, and 20 μg/mL) or 5-AzaDc after dissolving in a small amount of DMSO (0.1% *v*/*v* in cell culture medium).

### 2.3. Assay for Global DNA Methylation and DNMT Activity

Total genomic DNA was extracted from the cells and tumor tissues to analyze global DNA methylation. The global DNA methylation assay was performed following the manufacturer’s protocol. In this system, the methylated fraction of DNA is recognized by a 5mC antibody. This colorimetric assay kit quantified the amount of methylated DNA, which is proportional to the optical density. DNMT activity was determined in nuclear extracts following the kit protocol supplied by the manufacturer.

### 2.4. Analysis of 5mC in DNA following Dot-Blot Assay

Cells were treated with EGCG (0, 5, 10, and 20 μg/mL) for 6 days. Genomic DNA was isolated using the DNA isolation kit (Qiagen, MD), and dot-blot analysis was performed as described by Prasad et al. [20]. Briefly, genomic DNA (1 μg) was denatured and then blotted onto nitrocellulose membranes using Bio-Rad dot microfiltration apparatus (Bio-Rad Laboratories, Inc. Hercules, CA, USA). The membrane was fixed by baking for 30 min at 80 °C, then incubated with a primary antibody specific to 5-mC followed by a secondary HRP conjugated antibody. The membrane was developed on X-ray film using enhanced chemiluminescence detection reagents.

### 2.5. Analysis of Protein Levels Using Western Blotting

For protein expression analysis, cellular and nuclear fractions from HNSCC cell lines and tumor tissues were prepared using the cell fractionation kit purchased from Cell Signaling (#9038), following the manufacturer’s instructions. This cell fraction kit also contained protease inhibitor to avoid protein degradation, as described previously [21]. The 40–60 µg protein per sample was electrophoresed on 8–12% tris-glycine gels made in-house and then transferred onto nitrocellulose membranes. After blocking the membrane in 5% non-fat dry milk prepared in phosphate-buffered saline (PBS), the membranes were incubated with specific primary antibodies at 4 °C overnight, followed by an appropriate secondary horseradish peroxidase (HRP)-conjugated antibody. Specific proteins were visualized on X-ray film using an enhanced chemiluminescence reagent system. To confirm the equal loading of the protein on gels, β-actin was used for cytosolic fraction and histone H3 for nuclear fraction.

### 2.6. Tumor Xenograft Study in Balb/C Nude Mice

Female BALB/C nude mice (4–5 weeks of age) were purchased from the National Institute of Biologicals (Noida) and housed in the institutional animal resource facility. The mouse colony was maintained under a 12 h dark/12 h light cycle, a temperature of 24 ± 2 °C, and relative humidity of 50 ± 10%. The mice were given a control AIN76A diet with or without supplementation with EGCG (0.5%, *w*/*w*) and drinking water ad libitum throughout the experiment. The Institutional Animal Care and Use Committee, approved the animal protocol used in this study, under the animal protocol number 201809267. To determine the in vivo chemotherapeutic effect of EGCG on the growth potential of FaDu cells, 2 × 10^6^ cells (100 µL PBS) were injected subcutaneously in the right flank. One day after the cells’ injection, mice were divided randomly into two groups, 4 mice per group: (i) Control and (ii) EGCG treatment. Control mice were fed an AIN76A diet, while the other group was supplemented with 0.5% EGCG in their pellets throughout the experiment. AIN76A is the normal rodent chow and commercially available. The AIN76A diet maintained a balanced nutritional profile containing protein (18.1%), fat (5.1%), fiber (4.8%), and carbohydrate (65.2%). During the experiment, tumor size was measured weekly using calipers, and tumor volume was calculated by the ellipsoid formula (tumor volume = 1/2(length × width^2^) [22]. The experiment was terminated after 6 weeks, and tumor and body weights were recorded. At the end of the experiment, mice were euthanized with CO_2_ gas inhalation followed by cervical dislocation. Tumor tissues were collected and fractioned for protein estimation and histochemistry and stored at −20 °C until further analysis. The dietary dose of 0.5% EGCG is equivalent to 5 mg/mouse/day, assuming that a mouse of bodyweight 25.0 gm eats 5 g of chow [23]. Based on the mouse dose and formula by Nair et al., the human equivalent dose (HED) of 0.5% EGCG (*w*/*w*) would be approximately 189.18 mg/day [24].

### 2.7. Immunohistochemical Detection and Analysis

Paraffin-embedded tumor sections (5 µm thick) were deparaffinized, rehydrated, and an antigen retrieval procedure was carried out. Tissue sections were washed 3 times in PBS, blocked in 5% FBS, and incubated with specific primary antibodies overnight at 4 °C. After overnight incubation, sections were incubated with biotinylated secondary antibody followed by horseradish peroxidase-conjugated streptavidin. The sections were further incubated with 2, 4-diaminobenzidine (DAB) substrate and counterstained with hematoxylin and methyl green. The slides were photographed using an Olympus microscope.

### 2.8. Statistical Analysis

Data were evaluated for outliers and adherence to normal distribution, using GraphPad Prism software (San Diego, CA, USA), version 8.1. Statistical significance of normally and non-normally distributed data were assessed via one-way ANOVA and Tukey’s multiple comparison test, respectively, with α = 0.05.

## 3. Results

### 3.1. The Basal Levels of Global DNA Methylation in HNSCC Cell Lines

DNA methylation is an epigenetic alteration that frequently occurs during the early stages of carcinogenesis and affects cell division and differentiation. Therefore, we initially determined the basal levels of global DNA methylation in two HNSCC cell lines (SCC-1 and FaDu) and compared them with NHBE cells. After 3 days of cell culture (Figure 1A), significantly higher levels (F(2,6) = 74.50, *p* < 0.0001) of global DNA methylation (hypermethylation), 74% in SCC-1 (*p* < 0.005) and 178% in FaDu (*p* < 0.0001), were observed compared with the level of DNA methylation in NHBE cells. The changes in DNA hypermethylation were more profound after 6 days (F(2,6) = 246.1, *p* < 0.0001) as the levels of DNA hypermethylation were increased by 192% in SCC-1 (*p* < 0.005) and 375% in FaDu cells (*p* < 0.0001) compared with NHBE cells (Figure 1B). These results suggest that HNSCC cells underwent DNA hypermethylation. 

### 3.2. EGCG Treatment Inhibits DNA Hypermethylation in HNSCC Cell Lines

Next, to determine whether EGCG inhibits DNA hypermethylation in HNSCC cell lines, SCC-1 and FaDu cells were treated with EGCG dose- (0, 5, 10, and 20 µg/mL) and time-dependently for 3 and 6 days. After incubation with EGCG for the duration, cells were harvested and subjected to nuclear fractionation, and the levels of global DNA methylation were measured by ELISA. As shown in Figure 1C,D, The EGCG treatment to SCC-1 cells significantly decreased the global DNA hypermethylation, ranging from 20–43% after 3 days (F(3,4) = 24.99, *p* < 0.005) and 33–76% (F(3,4) = 78.81, *p* < 0.0005) after 6 days, compared with untreated cells. The inhibitory effect of EGCG on DNA hypermethylation was also similarly observed in the FaDu cells (Figure 1E,F). In comparison with the untreated group, the levels of DNA hypermethylation were greatly decreased in FaDu cells, up to 15–33% (F(3,4) = 48.39, *p* < 0.002) and 24–59% (F(3,4) = 84.99, *p* < 0.0004) after 3 days and 6 days, respectively. The inhibitory effect of EGCG on DNA hypermethylation was greater after 6 days of treatment. Our data suggest that inhibition of DNA hypermethylation might be a slow process, as the best effect of EGCG was observed after 6 days, therefore samples obtained after 6 days were analyzed for further verification.

### 3.3. EGCG Inhibits 5-mC Expression in SCC-1 and FaDu Cells

Further, to examine the dose-dependent effect of EGCG on DNA hypermethylation, SCC-1 and FaDu cells were treated for 6 days in a dose-dependent manner. The effect of EGCG on DNA hypermethylation in both cell lines was measured by dot-blot analysis using a 5-mc specific primary antibody. After 6 days of EGCG treatment, the expression of 5-mc was decreased in SCC-1 and FaDu cells (Figure 2A). The reduction of 5-mc expression was greater in FaDu cells than in the SCC-1 cells. Furthermore, the density of individual dot blots was assessed by densitometry analysis. The quantification of dot-blot analysis revealed that EGCG treatment significantly decreased 5-mc expression up to 39% in SCC-1 cells (*p* < 0.0001; Figure 2B) (F(3,4) = 56.23, *p* < 0.0001), and up to 60% in FaDu cells (*p* < 0.0001; (Figure 2C) (F(3,4) = 49.00, *p* < 0.0001), compared with the untreated group. The expression levels of 5-mc were further verified by immunostaining. As seen in Figure 2D, the numbers of brown nuclei (5-mc positive cells) were reduced following EGCG treatment both SCC-1 and FaDu cells compared with untreated cells. These results demonstrate that EGCG inhibits DNA hypermethylation in HNSCC cell lines.

### 3.4. EGCG Treatment Inhibits DNMT Activity, DNMT (DNMT1, DNMT3A, and DNMT3B) Protein Expression in SCC-1 and FaDu Cells

The process of adding a methyl group to the 5th position of cytosine, a conversion of cytosine to 5-methylcytosine, is regulated by DNA methyltransferases (DNMTs): DNMT1, DNMT3A, and DNMT3B [25,26]. Primarily, DNMT1 maintains DNA methylation, whereas DNMT3A and DNMT3B are responsible for de novo methylation [25,27]. Therefore, we next determined the effects of EGCG on DNMT activity and DNMT protein expression in SCC-1 and FaDu cells. After 3 and 6 days of treatment, EGCG significantly inhibited DNMT activity up to 54.38% (F(3,4) = 15.72, *p* = 0.0112) and 79.46% (F(3,4) = 33.31, *p* = 0.003) in SCC-1 cells compared with the untreated cohort, respectively (Figure 3A,B). The inhibitory effect of EGCG on DNMT activity was also observed in the FaDu cells (Figure 3C,D). The reduction of DNMT activity was greater in FaDu cells than in SCC-1 cells. The reduced expression of DNMT1, DNMT3A, and DNMT3B proteins was detected by Western blot in SCC-1 (left panel; Figure 3E) and FaDu (right panel; Figure 3E), compared with the untreated group. These results indicate that EGCG-mediated inhibition of DNA hypermethylation is associated with reduced DNMT activity and protein expression. As the effects of EGCG were comparable in both HNSCC cell lines, further mechanistic studies were conducted only with FaDu cells.

### 3.5. Effects of 5Aza-dc, a Potent Inhibitor of DNA Methylation, Alone or in Combination of EGCG, on DNA Hypermethylation and DNMT Activity

To confirm the inhibitory effect of EGCG on DNA hypermethylation, FaDu cells were further treated with 5Aza-dc alone and in combination with EGCG to assess the effect on DNA hypermethylation, DNMT activity, and DNMT protein expression. After 6 days, the dose and time-dependent application of 5Aza-dc to FaDu cells significantly reduced DNA hypermethylation up to 79.38% F(3,4) = 128.8, *p* = 0.0002) and DNMT activity up to 58.76% F(3,4) = 85.12, *p* = 0.0004) compared with the untreated cohort (Figure 4A,B).

To determine the inhibitory potential of EGCG on DNA hypermethylation, FaDu cells were treated with a combination of EGCG and 5Aza-dc for 6 days. For the purpose, an optimum dose of 5Aza-dc (5 µM) was selected. Our results demonstrated that combined treatment with EGCG and 5Aza-dc further increased the inhibitory potential of 5Aza-dc as observed by reduced DNA methylation and DNMT activity (Figure 4C,D). The combined treatment of EGCG and 5Aza-dc significantly decreased DNA hypermethylation up to 84.58% F(3,4) = 171.3, *p* = 0.0001) (Figure 4C) and DNMT activity up to 79.04% F(3,4) = 105.7, *p* = 0.0003) (Figure 4D) compared with the untreated group. The inhibitory potential of 5Aza-dc (5 µM) for DNA hypermethylation was increased up to 36.14% with EGCG (20 µg/mL) treatment compared with 5Aza-dc (5 µM) alone. Similar observations were observed for DNMT activity. The expression of DNMT1, DNMT3A, and DNMT3B proteins were also reduced after combined treatment, compared with the untreated group (Figure 4E). Altogether, these results suggest that EGCG has the potential to reduce DNA hypermethylation.

### 3.6. Effect of EGCG on DNA Demethylation in HNSCC Cells

Next, we assessed the effect of EGCG on 5-hydroxymethylcytosine (5hmC) expression. Upon DNA demethylation, 5-mc converts into 5hmC by ten-eleven translocation (TET) enzymes [21]. Therefore, to further confirm our results, the expression of 5hmC was detected by immunofluorescence staining in SCC-1 and FaDu cell lines after 6 days of treatment. As shown in Figure 5A, the expression of 5hmC was increased in both cell lines. Quantitative analysis of gray value intensity revealed significant upregulation of 5hmC expression (Figure 5B,C). Our results clearly demonstrate that EGCG blocks DNA hypermethylation by converting 5-mc into 5hmC.

### 3.7. EGCG Reactivates Tumor Suppressor Genes in HNSCC Cells

A family of cyclin-dependent kinase (CDK) inhibitor proteins (p16^INK4a^, p21^Waf1/Cip1^, and p27^Kip1^) act as tumor suppressors, and their inactivation contributes to human carcinogenesis [28]). Studies reported that DNA hypermethylation in these proteins’ promoter regions leads to suppression of their expression and eventual promotion of tumor growth and cell proliferation [29,30,31,32]. Previously, we showed that conversion of 5-mc into 5hmC reactivate silences tumor suppressors [21]. Next, we observed the effect of EGCG treatment on p16^INK4a^, p21^Waf1/Cip1^, and p27^Kip1^ expression in SCC-1 and FaDu cell lines (Figure 5D). After 6 days of EGCG treatment, the expression of these tumor suppressor proteins was increased in both cell lines in a dose-dependent manner. Increased expression of p16^INK4a^, p21^Waf1/Cip1^, and p27^Kip1^ suggests that EGCG reactivates tumor suppressor through DNA demethylation and may contribute to blocking cancer cells’ proliferation.

### 3.8. Growth Inhibitory Potential of EGCG on FaDu Xenograft in Nude Mice

As indicated by the in vitro studies demonstrating that EGCG treatment inhibits DNA methylation and reactivates tumor suppressor proteins, we sought to determine the growth inhibitory potential of EGCG using FaDu tumor xenograft in the immunosuppressed nude animal model. After 6 weeks of EGCG treatment, mice were euthanized and tumors were collected for histological and molecular analysis. As seen in Figure 6A, EGCG treatment blocked tumor growth in tumor-bearing mice compared with untreated mice. The tumor size in the EGCG-treated groups was not only smaller than in untreated mice, but was comparatively reduced by more than 95% (observed visually) in two mice. Furthermore, weekly measurements of tumor volume suggest that EGCG treatment blocked the tumor cells’ propagation from the beginning (week 1) (Figure 6B). Although there was a reduction in tumor volume, the average body weight was no different in EGCG-treated or non-EGCG-treated mice (Figure 6C). These data suggest no indication of toxicity for EGCG. The wet weight of the tumor per mouse was significantly reduced (0.65 ± 0.25 vs. 2.4 ± 0.31; *p* < 0.005) in the mice treated with EGCG compared with the untreated cohort (Figure 6B).

Next, we determined the effect of EGCG on DNA methylation and DNMT1, DNMT3a, and DNMT3b expression in tumor tissues. The expression of 5-mc (Figure 7A) and all three DNMT proteins (Figure 7B) was reduced in EGCG-treated mice, as reflected in less nuclear expression compared with non-EGCG-treated mice. The reduced expression of DNMT1, DNMT3a, and DNMT3b was also confirmed by Western blot analysis (Figure 7C). To verify the data obtained from cell culture studies, we also determined the effect of EGCG on tumor suppressor proteins by Western blotting in the tumors. Western blot analysis demonstrated that EGCG treatment reactivated silenced tumor suppressor proteins on p16^INK4a^, p21^Waf1/Cip1^, and p27^Kip1^ expression in the tumors, compared with tumors obtained from untreated mice (Figure 7D). These results further suggest that increased p16^INK4a^, p21^Waf1/Cip1^, and p27^Kip1^ expression blocks the cell cycle and ultimately contributes to the suppression of tumor growth in FaDu tumor xenografts.

## 4. Discussion

Head and neck cancers are heterogeneous in nature and occur at various anatomical sites, including in the oral cavity, tongue, pharynx, and larynx [33]. The incidence of HNSCC is continuously rising at an alarming rate up to a 30% increase in new cases anticipated by 2030 [34]. Epidemiological studies have demonstrated that cancers of the oral cavity and larynx are usually associated with excessive tobacco and alcohol consumption. In addition to tobacco and alcohol, environmental pollutants, and virus infections, mainly human papillomavirus (HPV) and Epstein–Barr Virus (EBV) may be implicated [33]. Cancer is a multifactorial phenomenon associated with various molecular changes at the cellular level. Genetic and epigenetic changes in the DNA contribute to altered gene expression and are induced by aging, mutagenic chemicals, nuclear radiation, ultraviolet radiation, oxygen radicals, and other factors [35]; genetic changes are irreversible, while epigenetic modifications, such as DNA hypermethylation, control gene expression and are reversible. Studies have shown that switching epigenetic changes can be beneficial for treating and managing human malignancies [36].

Due to its role in gene silencing mechanisms, DNA methylation has been extensively studied in human cancers [37]. Here, we studied the effects of EGCG on DNA hypermethylation in HNSCC, using in vitro and in vivo approaches. EGCG treatment significantly downregulated the DNA methylation and decreased DNMT1, DNMT 3A, and DNMT3B expression in SCC-1 and FaDu cells.

Although there are several DNA methylation inhibitors including 5Aza-dc, procainamide, hydralazine, and others, their ability to inhibit DNA methylation has been questioned and they may be weak [38]. Therefore, it is important to develop small molecules with greater potential for inhibiting DNA hypermethylation. The effectiveness of a drug as an epigenetic modifier depends on its effect at low doses, because high doses inhibit DNA synthesis and cause cells death, and lead to cytotoxicity. In the early clinical trials, 5Aza-dc was tested at high doses and exhibited cytotoxic effects [38]. Thus, the phase I/II sequence of drug testing failed in this case. Unfortunately, failure of clinical trials may not be limited to DNA methylation inhibitors, but is also applicable to other drugs. Our in vitro studies demonstrated that combined treatment of 5Aza-dc at low doses with active biomolecules such as EGCG will be beneficial, as natural compounds have minimal or no toxicity.

The cell cycle in normal cells is regulated by a set of proteins at various checkpoints. In cancer cells, the deregulated cell cycle results from the silencing or deactivation of various tumor suppressor genes [39]. Loss of tumor suppressor genes translates modification into resultant tumor growth. A family of cyclin-dependent kinase inhibitor proteins includes p16^INK4a^, p21^Waf1/Cip1^, and p27^Kip1^ [40]. The frequent inactivation of these tumor suppressor proteins is induced by homozygous deletion or promoter hypermethylation [28]. Our results demonstrate that EGCG treatment reactivates these silenced tumor suppressors and blocks FaDu tumor growth in nude mice.

Furthermore, other studies in skin cancer have shown that EGCG reactivated the silenced p16^INK4a^ and p21^Waf1/Cip1^ genes by decreasing DNA methylation and increasing histone acetylation [41]. Through inhibiting DNA hypermethylation, EGCG not only reactivates tumor suppressor genes, but also inhibits tumor promoter genes [42]. This study has demonstrated that EGCG has a promising role and great potential to regain the lost expression of tumor suppressor genes by inhibiting DNMT activities and DNA methylation in HNSCC cell lines in a time- and dose-dependent manner. However, further detailed studies are required in clinical settings to understand the complete mechanisms involved in the reversal of epigenetic modifications by green tea polyphenols.

## Figures and Tables

**Figure 1 biomedicines-11-00789-f001:**
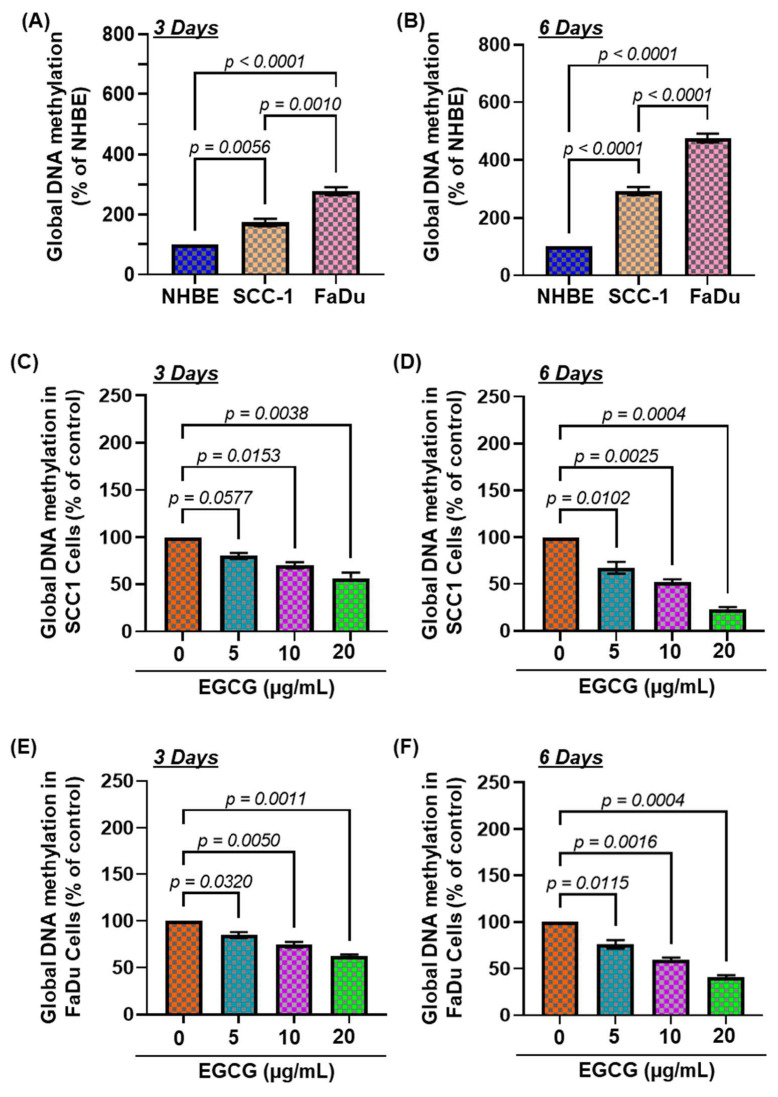
Effect of EGCG on global DNA methylation levels in head and neck squamous cell carcinoma cell lines. Comparative levels of global DNA methylation in NHBE, SCC-1, and FaDu cells. Data are presented in terms of percentage of NHBE, used as normal control cells. Significant difference of global DNA methylation level versus NHBE (**A**,**B**). Effect of EGCG on the global DNA methylation levels in SCC-1 (**C**,**D**) and FaDu (**E**,**F**) cells. Both HNSCC cell lines (SCC-1 and FaDu) were treated with various concentrations of EGCG (0, 5, 10, and 20 μg/mL) for 3 and 6 days, and the levels of global DNA methylation were determined using a global DNA methylation kit. Data are presented in terms of percentage of control (non-EGCG-treated) group, which was assigned a value of 100%, and as means ± S.E.M. n = 3.

**Figure 2 biomedicines-11-00789-f002:**
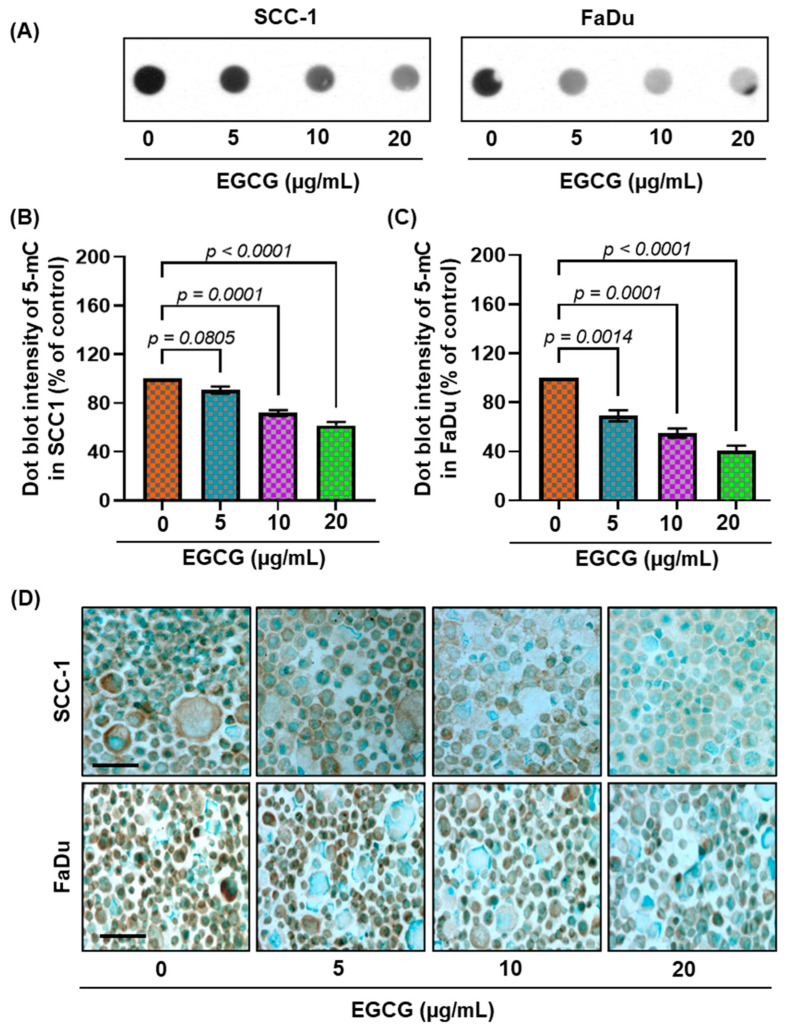
Treatment of SCC-1 and FaDu cells with EGCG decreases the levels of 5-mC in a dose- and time-dependent manner. Dot-blot analysis of 5-mc in DNA, extracted from various groups treated with or without EGCG in both cell lines (**A**). The intensity of individual dots was measured by densitometry in SCC-1 (**B**) and FaDu (**C**), and levels of 5-mc are presented in terms of relative density of dot blots as means ± S.E.M in different treatment groups, n = 3. Significant difference versus non-EGCG-treated controls. The effect of EGCG on DNA methylation in SCC-1 and FaDu cells was reconfirmed by immunostaining of 5-mc using an anti-5-mc antibody. Cells were counter stained with methylene green (**D**). Scale bar = 20 µM.

**Figure 3 biomedicines-11-00789-f003:**
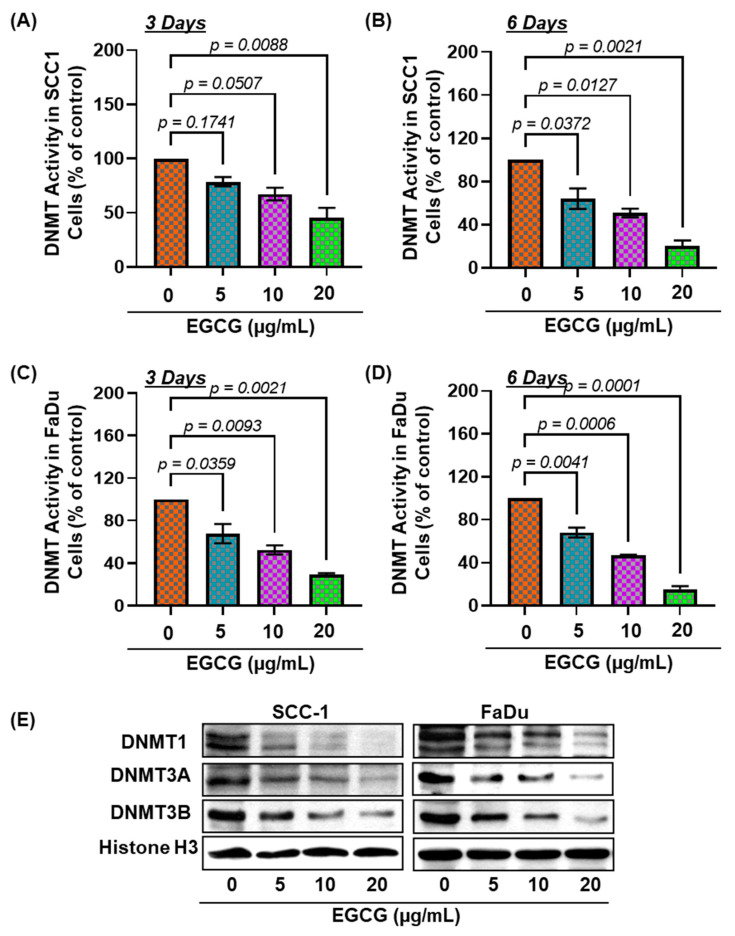
EGCG treatment inhibits DNA methyltransferases in HNSCC cells. Total DNMT activity in nuclear extracts of SCC-1 (**A**,**B**), and FaDu (**C**,**D**) was determined using the DNA methyltransferase activity assay kit. Data are presented in terms of percentage versus non- EGCG-treated controls, which were assigned a value of 100%, and as means ± S.E.M from three independent experiments. Significant differences were observed versus non-EGCG-treated controls. The levels of DNMT1, DNMT3a, and DNMT3b in nuclear lysates of SCC-1 and FaDu cells were determined using Western blot analysis after treating the cells with EGCG for 6 days (**E**).

**Figure 4 biomedicines-11-00789-f004:**
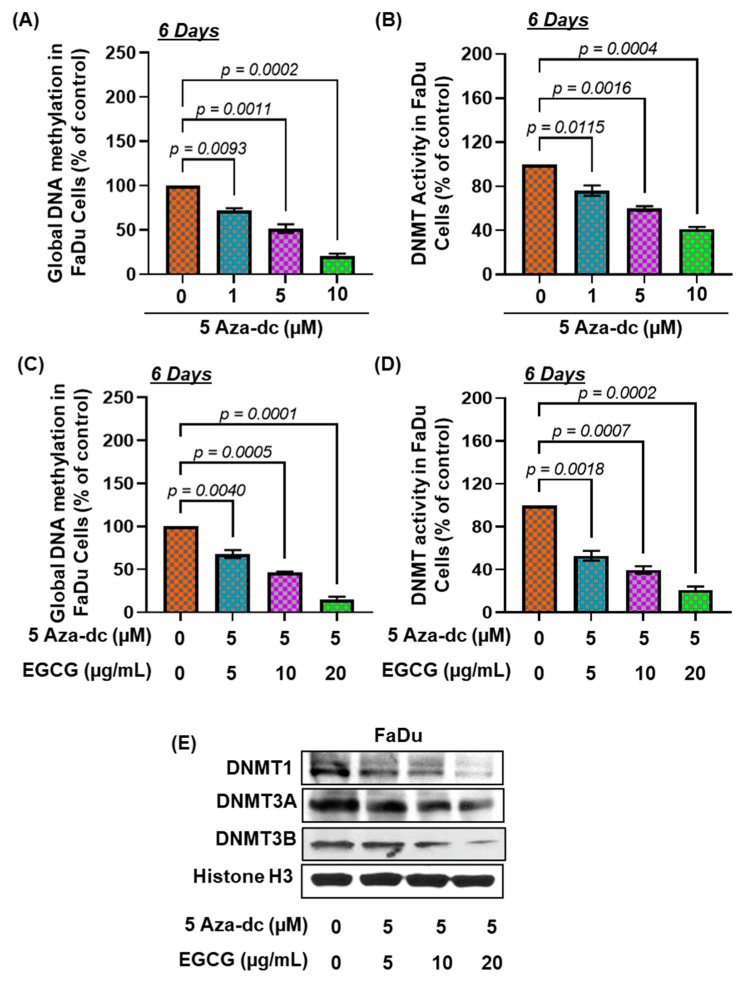
EGCG treatment synergistically modulates 5Aza-dc effects on DNA methylation and DNMT activity. FaDu cells were treated with 5Aza-dc in a dose-dependent manner for 6 days. The levels of global DNA methylation (**A**) and DNMT activity (**B**) in nuclear extracts were determined using global DNA methylation and DNA methyltransferase activity assay kit. The combined effect of EGCG and 5Aza-dc was observed on the expression level of global DNA methylation (**C**) and DNMT activity (**D**). Data are presented in terms of percentage versus non-EGCG-treated controls, which was assigned a value of 100%, and as means ± S.E.M from three independent experiments. A significant difference was observed versus non-EGCG-treated controls. The levels of DNMT1, DNMT3a, and DNMT3b in nuclear lysates of FaDu cells were determined using Western blot analysis after the combined treatment (**E**).

**Figure 5 biomedicines-11-00789-f005:**
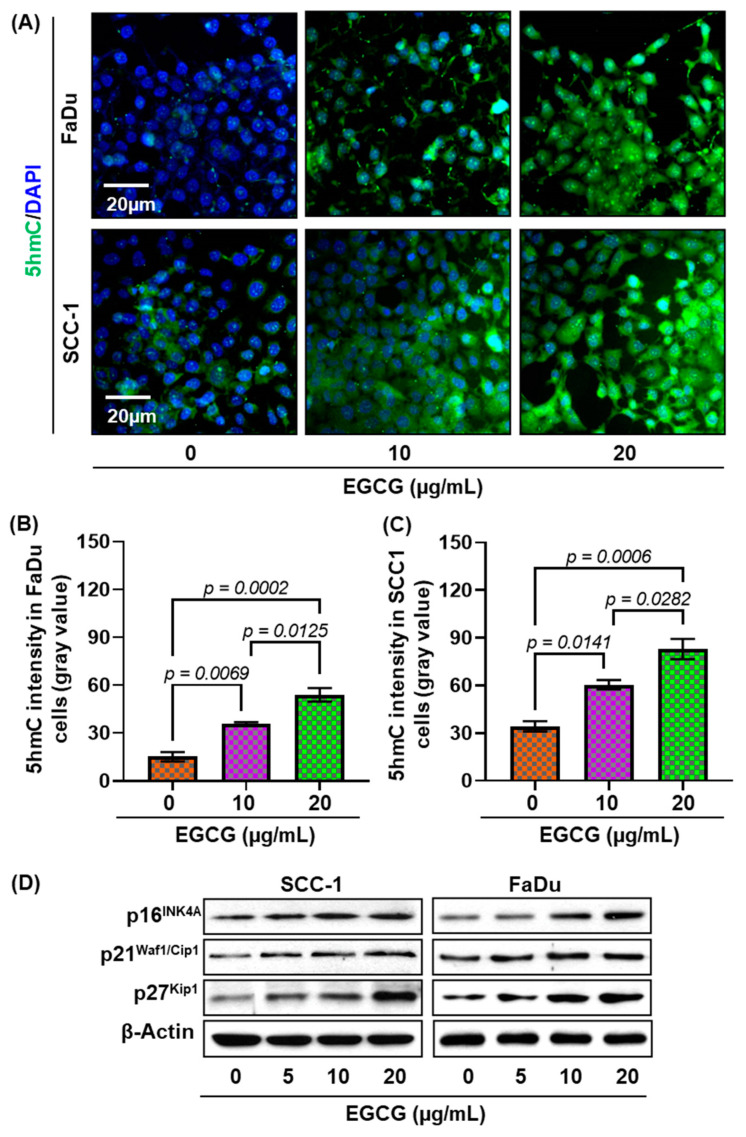
Effects of EGCG on DNA demethylation in HNSCC cells. SCC-1 and FaDu cells were treated with various concentrations of EGCG (10 and 20 μg/mL) for 6 days, and DNA demethylation was analyzed by immunostaining of 5-hydroxymethylcytosine (5hmC). Representative images of 5hmC staining (green) in FaDu and SCC-1 cells (**A**). The expression of 5hmC was quantified by ImageJ software and presented as mean gray value intensity ± S.E.M (**B,C**). **A** significant difference was observed versus non-EGCG-treated controls. EGCG treatment increased the expression of p16^INK4a^, p21^Waf1/Cip1^, and p27^Kip1^ compared with the untreated group (**D**).

**Figure 6 biomedicines-11-00789-f006:**
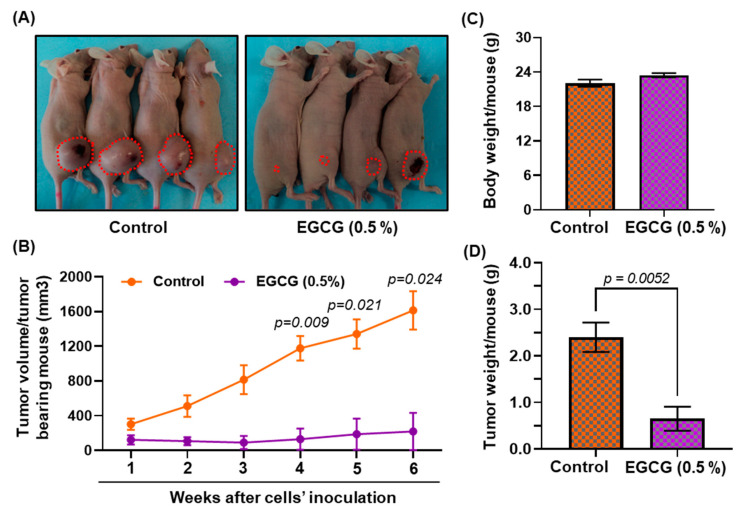
Dietary administration of EGCG inhibits the tumor growth of FaDu tumor xenograft in BALB/c nude mice. Mice were inoculated subcutaneously with 2 × 10^6^ cells (FaDu) on the right flank. Dietary administration of EGCG started one week before tumor cell inoculation. Representation of tumor bearing mice (**A**), tumor volume (**B**), body weight (**C**), and tumor weight (**D**). Data are presented as mean ± S.E.M. Statistical significance vs. non-EGCG-fed control group. n = 4.

**Figure 7 biomedicines-11-00789-f007:**
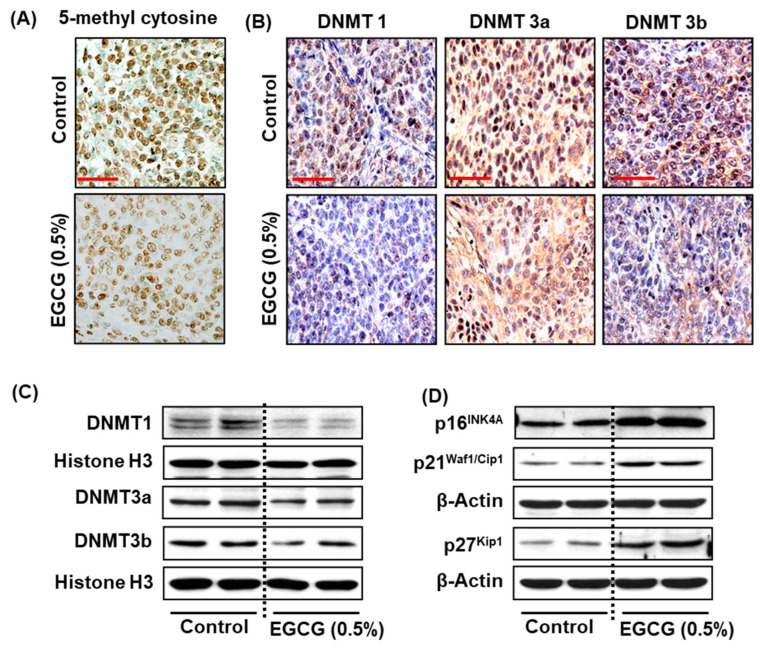
Effect of dietary administration of EGCG on epigenetic changes and tumor suppressor proteins in FaDu tumor xenografts in BALB/c nude mice. At the termination of the experiment, tumors from all groups were harvested and subjected to epigenetic changes. The effect of EGCG on global DNE methylation was assessed by immunostaining of 5-mc (**A**). The changes in DNMT1, DNMT3a, and DNMT3b protein expression were observed by immunostaining (**B**) and Western blot (**C**). EGCG administration increased the expression of p16^INK4a^, p21^Waf1/Cip1^, and p27^Kip1^ (**D**). Each column of the Western blot represents a pool of two tumor samples, obtained from two different mice. n = 4.

## Data Availability

The original data presented in the study are available on request from the corresponding author.

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
