# Peer review of "Epigallocatechin Gallate (EGCG), an Active Phenolic Compound of Green Tea, Inhibits Tumor Growth of Head and Neck Cancer Cells by Targeting DNA Hypermethylation"

_biomedicines, 2023, doi:10.3390/biomedicines11030789_

Round 1
Reviewer 1 Report
In this study, the author investigated the effect of EGCG on DNA hypermethylation and HNSCC growth.
They found that EGCG in FaDu and SCC-1 cells significantly inhibited global DNA hypermethylation by up to 70-80% after 6 days. The inhibition of DNA hypermethylation in HNSCC cells was further confirmed by the conversion of 5mC to 5hmC. They also found that EGCG in HNSCC cells significantly reduced DNMT activity to 60% in SCC-1 and 80% in FaDu cells. Furthermore, the protein levels of DNMT3a and DNMT3b were downregulated. During an animal tumor model study, the authors demonstrated that EGCG administration inhibited tumor growth of FaDu xenografts compared to EGCG-untreated controls associated with inhibition of DNA methylation, DNMTs, and reactivation of tumor suppressors silenced. In conclusion, the authors hypothesized that EGCG could be a DNA demethylating agent capable of reactivating epigenetically silenced
tumor suppressor factors.
The study is interesting, well-written and easy to follow.
My comments concern:
- the effects of dietary polyphenols (including EGCG) on DNA hypermethylation and tumor suppression are well known. Authors should carefully emphasize the purported novelty of their studies
- abbreviations must be written in full the first time they appear in the text
- an abbreviation list could be useful
- F and df should be included in the text
- nude female mice were used. Why? Did the authors check for "menstrual cycle" interference on tumor growth?
Reviewer 2 Report
In the present study the authors have investigated the effects of Epigallocatechin gallate (EGCG), a green tea extract, using in vitro models (cell lines) as well as in-vivo model. The authors have investigated the effects of the extract on DNA methylation, as well as tumor growth. They have shown that the extract was effective against tumor growth.
Their work is interesting, but unfortunately they should re-write large parts of their manuscript due to high similarity to other works.
Some further issues.
How were nuclei fractionated/extracted? What was the method used?
Please provide some details on the AIN76A diet.
DNA should not be abbreviated with punctuation.
Figure 6E, 6F the columns are not labelled.
The authors should highlight their findings. Further on, please comment on the dosages used in the in vitro and in vivo experimentation. What is the corresponding concentration in the human body? In the presented results, the higher concentrations are effective. To what concentrations correspond the proposed experimental concentrations? Simply put, how much tea should one have in order to reach therapeutic-related concentrations? What is the half-life of the extract in the human body? Please comment on the uses of the extract in its therapeutic and prevention uses in tumors.
Reviewer 3 Report
In fig.2, Inhibitors for DNMT1, 3a or 3b or siRNA should be used to see if effect of EGCG to inhibit methylation is dependent on these molecules.
In fig 3, methylation of specific promoters of p16, p21 and p27 should be shown by ChIP assay.
In fig 6, inhibitors for DNMT1, 3a, 3b p16, p21 or p27 should be examined to see if effect of EGCG to decrease tumor size is dependent on these molecules.
In fig 6, time dependent change of tumor diameter should be shown.
Round 2
Reviewer 1 Report
The authors properly addressed the comments of the reviewers
Reviewer 2 Report
The authors have addressed my previous comments.
Reviewer 3 Report
I do not have any more comments.